# The Need to Shift from Morphological to Structural Assessment for Carotid Plaque Vulnerability

**DOI:** 10.3390/biomedicines10123038

**Published:** 2022-11-24

**Authors:** Yuqiao Xiang, Xianjue Huang, Jessica Benitez Mendieta, Jiaqiu Wang, Phani Kumari Paritala, Thomas Lloyd, Zhiyong Li

**Affiliations:** 1School of Mechanical, Medical and Process Engineering, Queensland University of Technology, Brisbane, QLD 4000, Australia; 2School of Biological Science and Medical Engineering, Southeast University, Nanjing 210096, China; 3Department of Radiology, Princess Alexandra Hospital, Brisbane, QLD 4102, Australia

**Keywords:** luminal stenosis, carotid atherosclerosis, fluid–structure interaction, plaque vulnerability, structural assessment

## Abstract

Degree of luminal stenosis is generally considered to be an important indicator for judging the risk of atherosclerosis burden. However, patients with the same or similar degree of stenosis may have significant differences in plaque morphology and biomechanical factors. This study investigated three patients with carotid atherosclerosis within a similar range of stenosis. Using our developed fluid–structure interaction (FSI) modelling method, this study analyzed and compared the morphological and biomechanical parameters of the three patients. Although their degrees of carotid stenosis were similar, the plaque components showed a significant difference. The distribution range of time-averaged wall shear stress (TAWSS) of patient 2 was wider than that of patient 1 and patient 3. Patient 2 also had a much smaller plaque stress compared to the other two patients. There were significant differences in TAWSS and plaque stresses among three patients. This study suggests that plaque vulnerability is not determined by a single morphological factor, but rather by the combined structure. It is necessary to transform the morphological assessment into a structural assessment of the risk of plaque rupture.

## 1. Introduction

Atherosclerosis is one of the main underlying causes of cardiovascular disease (CVD) [1]. Morphological features and biomechanical environment have been found to play an important role in the progression and rupture of atherosclerotic plaques [2,3,4]. The degree of luminal stenosis is typically considered an important indicator for judging the risk of atherosclerosis burden. Moreover, plaque morphology has been recognized to be critically important to better assess plaque vulnerability [5,6,7]. Biomechanical modelling can provide the mechanical interaction between blood flow and plaque, which has been reported recently to be a functional tool to evaluate the risk of plaque rupture [8]. However, it remains unknown if there are significant differences in plaque morphology and biomechanical factors among patients with the same or similar degrees of stenosis. Thus, improved knowledge is needed to understand why we should shift from morphological to functional risk assessment for plaque vulnerability.

An increasing number of studies have suggested that hemodynamic factors play a key role in the initiation, development, and progression of atherosclerosis [9,10,11]. The development and progression of atherosclerotic plaque is usually found in areas with complex flow patterns, such as the carotid artery bifurcation [12,13,14]. Mathematical and numerical models have been developed to study the mechanobiological processes of plaque progression and identify possible prevention and treatment strategies [15,16,17,18]. Local hemodynamic forces have a profound impact on the formation and progression of plaque in the carotid area [19,20]. For this reason, the importance of wall shear stress (WSS) has been widely explored to identify its relationship with the progression and rupture of carotid plaque. Low WSS induces endothelial cell phenotypic transition and an atherogenic gene expression [21,22]. Conversely, high WSS is associated with plaque instability, which can further lead to rupture, platelet aggregation, and atherosclerosis progression [23,24].

Computational fluid dynamics (CFD), a numerical field of fluid dynamics, has received increasing attention as a tool for the study of cardiovascular hemodynamics [25,26,27]. The interaction between the blood domain and deforming arterial walls is not typically considered, and CFD simulations are usually restricted only to the blood flow analysis. However, fluid–structure interaction (FSI) models combining CFD with structural finite element analysis (FEA) have been provided to give a more accurate estimation of the real vascular system and have been used to assess both fluid dynamic and structural behaviors in human atherosclerotic carotid plaques [28,29,30].

Using the FSI modelling approach developed over the last few years, we aimed to investigate the difference in the functional (biomechanical) factors in patients who have a similar luminal stenosis, thus providing an example to illustrate why there is a need to shift from morphological plaque assessment to a functional evaluation of plaque rupture risk. This study analyzed and compared the biomechanical factors of three patients with a similar carotid stenosis. First, different components of the carotid plaque were segmented based on magnetic resonance imaging (MRI), including the lumen, outer walls, lipids, and calcification. By deriving the coordinates of the atherosclerotic plaque tissue, the material properties of different tissues were directly mapped to each finite element corresponding to the coordinates. Finally, a two-way fluid–structure coupling calculation was performed on the model, and patient-specific flow and pressure conditions were used in the calculation. A variety of biomechanical risk factors of the three patients were analyzed and compared.

## 2. Materials and Methods

### 2.1. Patients and MRI

Three patients with similar carotid stenoses who were planned for carotid endarterectomy (CEA) were included in this study. The degrees of stenosis were calculated according to the North American Symptomatic Carotid Endarterectomy Trial (NASCET) method [31]. Patient demographics are provided in Table 1. The clinical study was conducted at the Princess Alexandra Hospital (PAH) in Brisbane, Australia. This study was approved by the Metro South Human Research Ethics Committee (HREC/17/QPAH/181) and patient consent forms were obtained. This study was performed in accordance with the guidelines of the Declaration of Helsinki.

Before performing CEA, multi-contrast MRI was performed at the carotid bifurcation of each patient on a 3T whole body system (Magnetom Prisma, Siemens, Malvern, PA, USA). Four contrast-weighted imaging sequences (including T1-weighted (T1W), T2-weighted (T2W), proton-density-weighted (PDW), and time-of-flight (TOF)) were obtained to allow for the identification of the different plaque components. Additionally, 2D electrocardiogram (ECG)-gated phase-contrast MRI (PC-MRI) images at approximately 40 equidistant time frames in the cardiac cycle were acquired to record the mass flow profile at three different locations, including common carotid artery (CCA), maximum stenotic region, and internal carotid artery (ICA).

### 2.2. Segmentation and Reconstruction

The lumen centerline was determined as the least-cost path between user-defined seed points in the CCA, ICA, and external carotid artery (ECA). Two least-cost paths were computed using the Dijkstra algorithm, one between ICA and CCA, and the other between ECA and CCA. Subsequently, initial contours were generated according to the positions of the two centerlines, and an improved active contour model [32] was used to detect the lumen. The outer wall was segmented using a path-tracing method based on the circle model [33,34]. After detecting the lumens, the tracking algorithm was applied to radial profile gradients to detect the outer wall of the vessel. In brief, a notion of the local minimal path that aimed at restricting the tracking to given orientations and distance was pioneered. To segment the plaque components, a k-means clustering algorithm [35,36] was used to perform tissue clustering between the inner and outer walls. The accuracy of the method and the segmentation result have been evaluated by experienced radiologists at the Princess Alexandra Hospital.

### 2.3. Material Mapping Method

The solid geometric models were meshed with the proximity and curvature size function in Ansys Meshing (version 19.0, ANSYS, Canonsburg, PA, USA). An in-house MATLAB code was designed to generate the 3-dimensional coordinates of segmented artery components, including arterial walls, lipids, and calcification. By deriving the coordinates of the atherosclerotic plaque tissue, the material properties of different tissues were mapped to each element in the meshed structure corresponding to the coordinate data. The linear elastic material properties (Young’s modulus) were assigned to each component (arterial tissue, 0.6 MPa; calcification, 10 MPa; lipids, 0.02 MPa), Poisson’s ratio for each component was set as 0.48 [37,38,39]. An interpolation method was applied between the material properties of different components, which provided a transitional region between plaque components and the arterial wall.

### 2.4. FSI Computational Model

Blood flow was assumed to be laminar, Newtonian, viscous, and incompressible. The viscosity and density of blood were assigned as 0.00345 Pa·s and 1050 kg/m^3^, respectively [40]. The fluid domain was meshed with tetrahedral elements with a size of 0.3 mm, as a result of mesh independence testing. The no-slip boundary condition was assumed to be the fluid domain, which was widely used in simulations of stenosed carotid bifurcations [41,42]. Patient-specific time-dependent mass flow rate waveforms acquired from PC-MRI at CCA were set as the inlet boundary condition. Figure 1 shows the inlet boundary conditions used for the three patients. Based on the mass flow rate profile, the pressure profile (as outlet boundary condition at ICA and ECA) was scaled within the range of measured diastolic and systolic pressure values of each patient. In the structural participant, fixed supports were added to the side edges of the geometries at three ends (CCA, ICA, and ECA). The inner surface of the arterial wall, which is in contact with the lumen, was set as fluid–solid interface for data transfer. The coupled FSI plaque models were solved by a commercial finite element package ANSYS Workbench (version 19.0, 2019, ANSYS Inc., Canonsburg, PA, USA). The system coupling component was used for the allowance of pressure/force data transfer and displacement information between the two participants (fluid and structural) in FSI. No external load was applied to the vessel wall, and the blood vessel only received the pressure transferred from the fluid participant [38]. The calculation time step was set to 0.01 s for two-way FSI simulation. For the data transfer at each time step, linear ramping data was transferred between structural and CFD in the minimum iteration number, which was five iterations. All simulations were achieved with the same machine (Intel(R) Xeon(R) CPU E5-2680 v4 @2.40 GHz, 56 cores; RAM: 128 GB), while the average computational time was 60 h for one simulation.

### 2.5. Analysis of FSI Result

Time-averaged wall shear stress (TAWSS) and oscillatory shear index (OSI) were used to evaluate the flow behavior in the carotid bifurcation. These parameters were defined as:(1)$$TAWSS=\frac{1}{T}\;\int_{0}^{T}\left|\tau_{w}\right|dt$$
(2)$$OSI=\frac{1}{2}\;\left(1-\frac{\left|\int_{0}^{T}\tau_{W}dt\right|}{\int_{0}^{T}\left|\tau_{W}\right|dt}\right)$$
where $\tau_{W}$ is the instantaneous wall shear stress and T is the cardiac cycle period. The maximum principal stress (Stress-P1) values were used to evaluate the stress distribution within the plaque. Data analysis and visualization were performed with R software (version 4.2.0).

## 3. Results

### 3.1. Carotid Plaque Morphology

Although their degrees of carotid stenosis were similar, the plaque components of the segmented carotid bifurcations, especially the calcification content, showed a significant difference. Figure 2a shows the reconstruction result of the carotid bifurcations of three patients, where Figure 2b shows the example of segmented contour plots on T1W images of patient 1.

Table 2 shows the carotid plaque morphology of three patients. A large calcification volume was found in patient 2, which was 22.4 times the volume of patient 1 and 7.4 times the volume of patient 3. The highest value of lipid volume was found in patient 3, which was 5.5 times the volume of patient 1 and 4.87 times the volume of patient 2, respectively. The thinnest fibrous cap thickness of patient 1 and patient 3 was 0.729 mm and 0.676 mm, respectively. Patient 2 had a relatively thicker fibrous cap, of which the thinnest thickness was 1.186 mm.

### 3.2. PC-MRI and CFD Comparison

A qualitative comparison between the PC-MRI measurements and CFD simulation results was performed for all three patients. Figure 3 shows the comparison of patient 1, where the axial velocity at systole in the same cross-section at the ICA was selected for comparison. In all three cases, the overall flow pattern was found to be similar. However, CFD results were found to overestimate the maximal velocity values. Compared with PC-MRI measurements, the maximum velocity values from CFD simulation were found to be 26.6% higher for patient 1, 23.9% higher for patient 2, and 18.6% higher for patient 3.

### 3.3. WSS-Based Descriptors

High WSS was found at the stenotic location. Figure 4 shows the WSS distribution for the three patients at peak systolic phase. A region of interest was defined for each patient, starting from the bifurcation apex and covering the stenotic area at the ICA with a length of 17 mm. The WSS range of patient 2 was found to be higher than the WSS value of patient 1 and patient 3.

Figure 5 shows the TAWSS and OSI distribution for the three patients. High values of TAWSS were found in the area near the stenotic region (Figure 5a). Also, the high TAWSS value of each carotid artery model could be found at the bifurcation apex and the significant narrowing of the ICA branch. For OSI, a high value of OSI could be found at either the CCA branch near the bifurcation apex or the ICA branch downstream of the stenosis, as marked with red circles (Figure 5b).

Figure 6 shows the distribution of TAWSS values at the region of interest (as highlighted in Figure 4) of the three patients. A relatively lower TAWSS value was found in patient 1 and 3 than that in patient 2. For the three patients, the mean value and the standard deviation of TAWSS at the region of interest were evaluated (patient 1, 5.45 ± 6.44 Pa; patient 2, 14.62 ± 13.72 Pa; patient 3, 6.99 ± 6.34 Pa). Significant differences in the TAWSS were observed where the range of TAWSS of patient 2 was wider than that of patient 1 and 3.

### 3.4. Plaque Stress

The maximum principal stress (Stress-P1) values were used to evaluate the stress distribution within each plaque. High Stress-P1 value could be found near the plaque area where the fibrous cap was thin. In addition, high Stress-P1 was found in the region near the lumen with a large curvature.

At peak systole, the stress distribution and the Stress-P1 and the corresponding zoomed views (excluding the plaque structure) are shown in Figure 7a. The calculated maximum Stress-P1 values for the three patients were 133.1 kPa, 116.9 kPa, and 127.4 kPa, respectively. For the analysis of plaque stress, a region of interest was defined for each patient, which selected the plane with the local Stress-P1 as the central plane and covered a length of 13 mm. The mean value and the standard deviation of stress values in the regions of interest were 22.221 ± 16.579 kPa for patient 1, 13.098 ± 20.378 kPa for patient 2, and 21.075 ± 17.391 kPa for patient 3. Figure 7b shows the distribution of the Stress-P1 in the ROI. There were significant differences in plaque stress among the three patients. Patient 2 was found to have a much smaller plaque stress compared to the other two patients.

## 4. Discussion

The area [43], shape [39], and location [44] of calcification were closely related to the stability of the plaque. Large areas of calcification provided stability to the plaque, making it harder and less likely to rupture [45,46]. Different patterns of fiber organization around the calcifications also had effects on the stress distribution in plaque tissue [47]. The interaction effects produced by calcifications and lipids can be complex and significant [48]. Differences in the volume and location of calcifications were observed among patient 1, 2, and 3. Patient 2 had a larger calcification volume than patient 1 and patient 3. The location of calcification in relation to lipid of patient 2 also showed obvious difference compared to patient 1 and patient 3. These variances in calcification may contribute to the differences in stress distribution.

Plaque rupture risk stratification has been discussed for approximately two decades. Growing evidence has shown that luminal stenosis alone is not sufficient for assessing the atheroma burden [49], and plaque morphology [50] has been widely considered to be crucial for determining whether a plaque is at high risk of rupture, particularly for patients with a moderate stenosis. More recently, image-based biomechanical modelling has been developed to provide hemodynamic (such as WSS-based descriptors) and wall stress parameters, which has shown its potential for use as a functional tool to better assess plaque structural stability. However, there is still a lack of a quantitative evaluation of plaque vulnerability.

Our results are consistent with those of previous studies [30,43] that highlighted the crucial role of considering both the morphology and the mechanical properties of different plaque components in addition to the degree of carotid stenosis in determining plaque vulnerability. Although the American Heart Association (AHA) has summarized a classification of atherosclerosis [51], the understanding of why such a morphological classification is linked with plaque vulnerability has not been fully illustrated. From the perspective of structural evaluation, plaque rupture is considered to be a structural failure when the plaque cannot resist the hemodynamic blood pressure and shear stress imposed on it [52,53]. Quantification of plaque morphology provided basic information for the structural assessment of plaque vulnerability [54]. For patients with similar luminal stenoses, significant differences may be found in plaque morphology. In our study, the fibrous cap thickness of three patients showed no significant difference. However, the lipid core and calcification volume showed significant differences. Significant differences were also found in TAWSS and plaque stresses among the three patients, where the distribution of TAWSS and Stress-P1 of patient 1 showed a similar trend to that of patient 3. Our result showed that the TAWSS distribution range of patient 2 was wider than that of the other patients, where the mean TAWSS value was 2.7 times the value of patient 1 and 2.1 times the value of patient 3. Patient 2 also had a smaller plaque stress distribution, when compared with patient 1 and 3. The maximum Stress-P1 value of patient 2 was 13.8% smaller than that of patient 1, and 8.9% smaller than that of patient 3. For local plaque stress distribution in the selected plane, patient 2 was found to have the smallest mean stress value among the three patients, which was 69.7% smaller than that of patient 1 and 60.9% smaller than that of patient 3. Between patient 1 and patient 2, the major difference in morphological factor was found in the calcification volume. However, between patient 1 and patient 3, only the lipid volume showed a significant difference. These may suggest that the functional assessment of plaque vulnerability is not dependent on a single morphological parameter, but rather a combined structure. Biomechanical parameters described in this study (such as TAWSS, OSI, and maximum principal stress) can well reflect the differences and changes in terms of the hemodynamic shear stress and plaque stress of the carotid artery structure.

Therefore, there is a need to shift morphological plaque assessment to a structural evaluation of plaque rupture risk, where the true vulnerability is assessed by integrating the plaque structure with its surroundings.

This study had some limitations. In this study, blood flow was assumed to be laminar. Blood flow through stenotic carotid bifurcations may experience transition to turbulence, the presence of which may have some effects on plaque rupture [55,56]. The material properties were based on previous studies, and linear elastic behavior was assumed for arterial tissue, calcification, and lipids. The estimation of patient-specific plaque material properties can be further improved through non-invasive techniques, such as ultrasound elastography and in vivo MRI [57,58,59]. Another limitation was the limited number of patients. Further large-scale studies are needed to provide more statistically reliable information.

## 5. Conclusions

In this study, three patients with carotid atherosclerosis within a similar range of stenosis were studied. Through the developed FSI modelling approach, the biomechanical risks of three patients were compared. Although their degrees of carotid stenosis were similar, the plaque components, distribution of TAWSS, and Stress-P1 showed significant differences.

This study illustrated that the plaque structural stability was significantly different among three patients, although their degree of stenosis was at a similar level. Therefore, it suggests that plaque vulnerability is not determined by a single morphological factor but rather by a combined structure. There is a need to shift from morphological to structural assessment of plaque rupture risk.

## Figures and Tables

**Figure 1 biomedicines-10-03038-f001:**
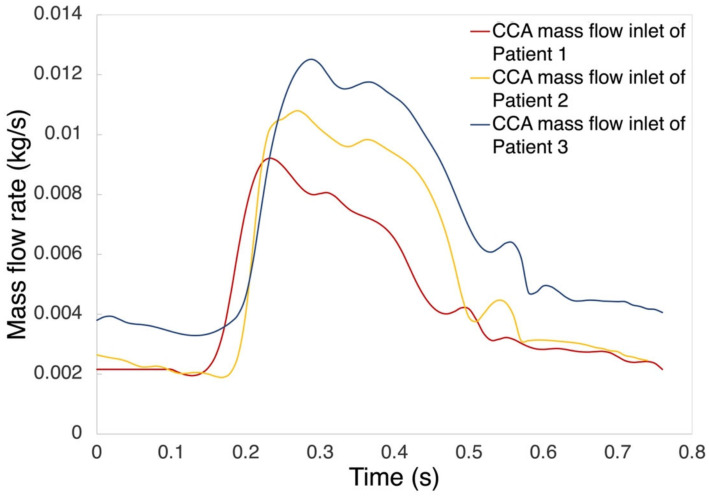
Inlet boundary conditions used for three patients.

**Figure 2 biomedicines-10-03038-f002:**
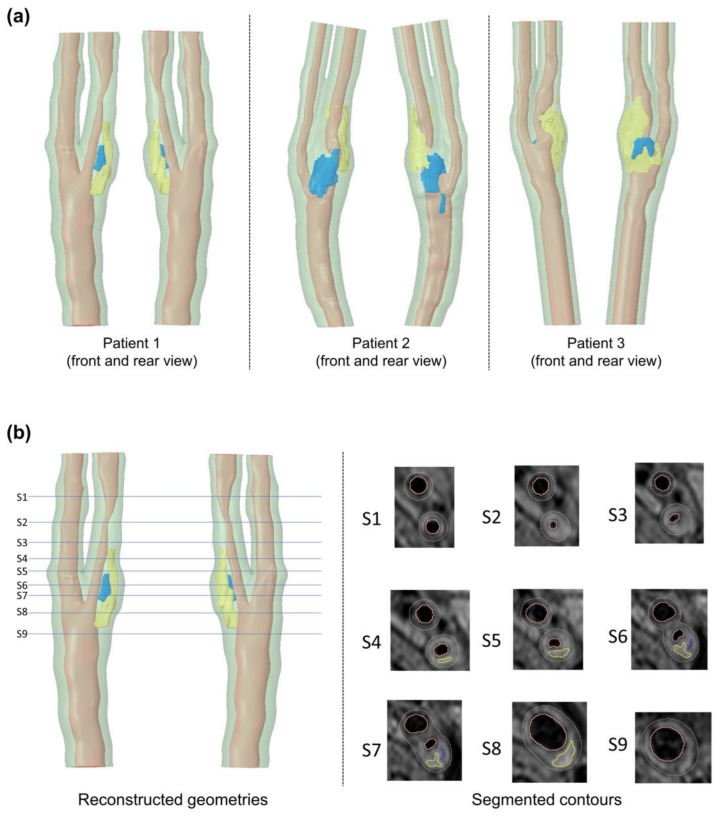
Reconstructed carotid bifurcation models showing: (**a**) carotid bifurcation geometries of three patients, and (**b**) segmented contour plots on T1W images of patient 1 (light red = lumen, green = arterial tissue, yellow = lipids, blue = calcification).

**Figure 3 biomedicines-10-03038-f003:**
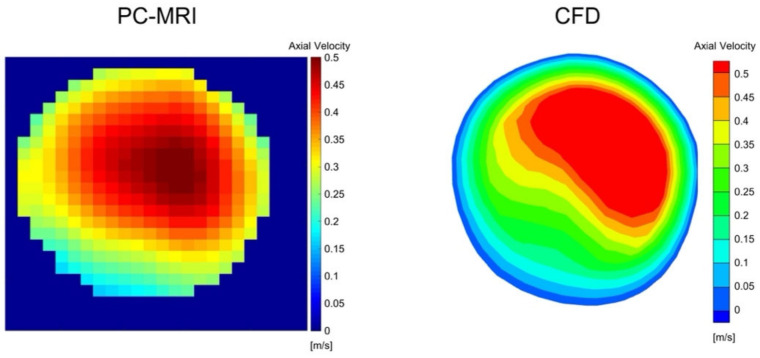
Velocity comparison between PC-MRI measurements and CFD simulations at the location of ICA of patient 1.

**Figure 4 biomedicines-10-03038-f004:**
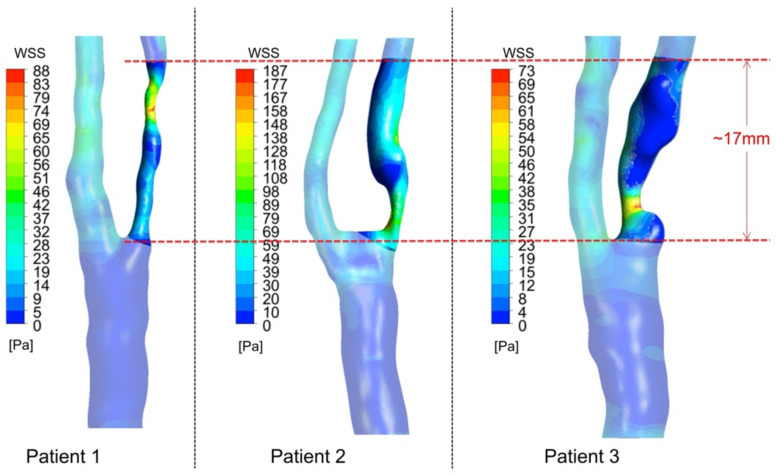
Result of wall shear stress (WSS) distribution at peak systole.

**Figure 5 biomedicines-10-03038-f005:**
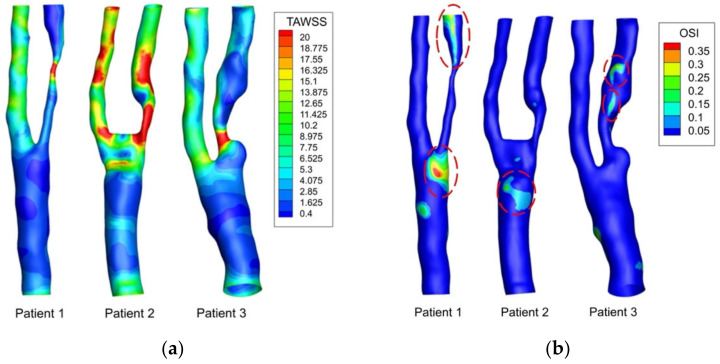
Result of (**a**) time-averaged wall shear stress (TAWSS) and (**b**) oscillatory shear index (OSI) distribution for three patients.

**Figure 6 biomedicines-10-03038-f006:**
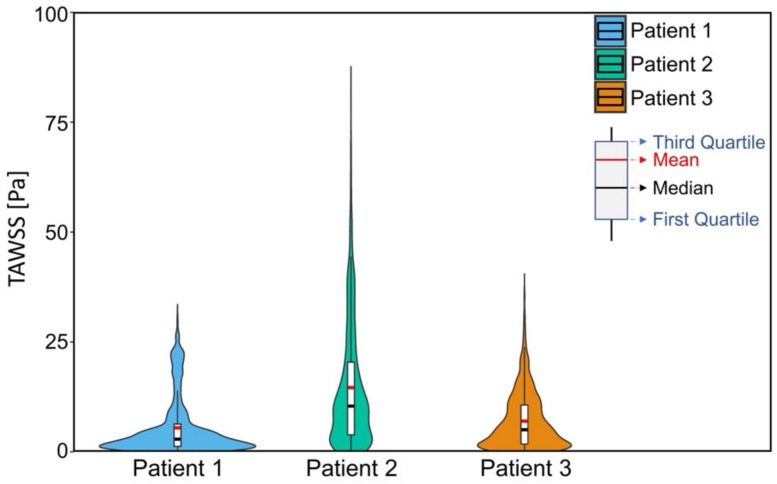
Violin plot of TAWSS values at region of interest in three patients. For each patient, the median value is shown in black-colored line and the mean value is shown in red-colored line.

**Figure 7 biomedicines-10-03038-f007:**
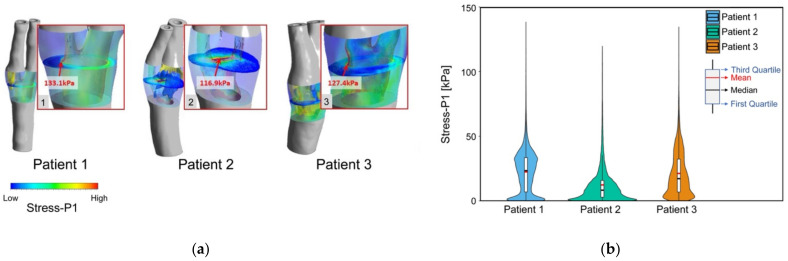
Result of Stress-P1 of three patients at peak systole. Images (**a**) (1–3) show contour maps of Stress-P1 for the selected region of interest and corresponding zoomed views excluding the plaque structure, with arrows showing the local maximum Stress-P1 in plaque; (**b**) violin plot of the Stress-P1 in the region of interest of each patient.

**Table 1 biomedicines-10-03038-t001:** Patient demographics.

Patient ID	Patient 1	Patient 2	Patient 3
Age	42	77	70
Sex	Male	Female	Male
Degree of stenosis	81%	83%	82%
BMI (kg/m^2^)	21.2	16	22.3
Hypertension	Yes	Yes	Yes
Hypercholesterolemia	Yes	No	Yes
Diabetes Mellitus	No	No	No
Smoking	Former	Current	Current

**Table 2 biomedicines-10-03038-t002:** Carotid plaque morphology differences among three patients.

Patient ID	Patient 1	Patient 2	Patient 3
Calcification volume (mm^3^)	3.44	77.38	10.43
Lipid volume (mm^3^)	42.79	48.59	236.79
Thinnest fibrous cap thickness (mm)	0.729	1.186	0.676

## Data Availability

The data presented in this study are available on request from the corresponding author.

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
