# Peer review of "The Need to Shift from Morphological to Structural Assessment for Carotid Plaque Vulnerability"

_biomedicines, 2022, doi:10.3390/biomedicines10123038_

Round 1

Reviewer 1 Report

Three atherosclerotic carotid arteries with similar stenosis diameter were modeled in ANSYS to measure the shear and wall stresses. The paper claims that the mechanical state would play a significant role compared to the angiographic measures such as minimum lumen diameter (stenosis). The topic bears significance, but the presentation in terms of quantification is very poor. There are multiple comments to address:

-        Further evidence for the accuracy of segmentation and geometry generation is needed. At least, the qualitative comparison of the cross-sectional images and masks.

-        Authors should justify the material model selection, i. e. linear elastic, especially for the lipid and healthy tissue. Majority of published work support to model atherosclerotic arteries using hyper elastic models.

-        Hemodynamic modeling of highly stenosed small arteries might require non-Newtonian assumption, unless the shear strain analysis proves otherwise. Please make a comment on such an evaluation.

-        Mesh independence was not discussed, nor was the computational time and cost. Methods part is too brief to be reproducible.

-        How do the authors could justify that the stenosis axial distance from the site of bifurcation would not affect their hemodynamics? The lumen and the location of stenosis looks pretty different in Patient 1 than the others.

-        What is the region of interest in Figure 5? The whole modelled domain or what was shown in Fig3?

-        Section 3.4 needs to be moved to the methods part or the beginning of results to be used as the validation of simulations. Authors then can present the results arguing that the simulations are accurate enough to proceed.

-        What is the first paragraph of the discussions? It looks more like a comment from a former review.

-        The authors should standardize the color bar in Figure 7 for a more fair comparison of PC-MRI vs CFD.

-        The authors in the discussion rightly highlight the growing importance of calcium in mechanical stability through factors such as area, shape, and location. We highly suggest discussing other recent literature discussing [1] the importance of incircling fiber patterns around the calcium and [2] the importance of understanding that calcium can both increase and decrease local wall stress, depending on where the stress is measured relative to the calcium.

Ref [1]: Gijsen, Frank JH, et al. "Morphometric and mechanical analyses of calcifications and fibrous plaque tissue in carotid arteries for plaque rupture risk assessment." IEEE Transactions on Biomedical Engineering 68.4 (2020): 1429-1438.

Ref [2]: Kadry, Karim, et al. "A platform for high-fidelity patient-specific structural modelling of atherosclerotic arteries: from intravascular imaging to three-dimensional stress distributions." Journal of the Royal Society Interface 18.182 (2021): 20210436

-        Is there a better way of quantifying the differences between patient 1 and 3 in terms of spatial distribution of calcium and lipid? Paragraph starting at line 229 is too brief and rough. Same is the paragraph starting at 245.

-        Some typos: e.g. Line 237: “is not sufficient for assess”

Reviewer 2 Report

The paper deals with a FSI computational study in three stenotic carotids in presence of different plaque typologies. The topic and the results are interesting for the journal, however some issue should be carefully addressed by the authors.

MAJOR REMARKS:

1. Line 41: More classical references on this topic should be added, e.g.

Ku, D., Giddens, D., Zarins, C., and Glagov, S., 1985, “Pulsatile Flow and Atherosclerosis in the Human Carotid Bifurcation: Positive Correlation Between
Plaque Location and Low and Oscillating Shear Stress,” Arterioscler., Thromb.,
Vasc. Biol., 5(3), pp. 293–302.

2. Line 45: Here the authors could introduce a short comment on differential models coupled to blood dynamics to account for the plaque progression and add some references, e.g.

Yang, Y., Jager, M., Neuss-Radu, W., Richter, T.: Mathematical modeling and simulation of the evolution of plaques in blood vessels. J. Math. Biol. 72, 973–996 (2015)

Thon, M., Hemmler, A., Glinzer, A., Mayr, M., Wildgruber, M., Zernecke-Madsen, A., Gee, M.: A multiphysics approach for modeling early atherosclerosis. Biomech. Model. Mechanobiol. 17, 617–644 (2017)

Pozzi S., Redaelli A., Vergara C., Votta E., Zunino P., Mathematical and numerical modeling of atherosclerotic plaque progression based on fluid-structure interaction. Journal of Mathematical Fluid Mechanics, 23, 74, 2021

3. References about previous FSI studies for carotids with plaque are completely absent (if I am not wrong). The authors should cite some works on the subject, e.g.

Tao, X., Gao, P., Jing, L., Lin, Y., and Sui, B., 2015, “Subject-Specific Fully-
Coupled and One-Way Fluid-Structure Interaction Models for Modeling of
Carotid Atherosclerotic Plaques in Humans,” Med. Sci. Monit., 21, pp.
3279–3290.

Tang, D., Teng, Z., Canton, G., Yang, C., Ferguson, M., Huang, X., Zheng, J.,
Woodard, P., and Yuan, C., 2009, “Sites of Rupture in Human Atherosclerotic
Carotid Plaques Are Associated With High Structural Stresses an In Vivo MRIBased 3D Fluid-Structure Interaction Study,” Stroke, 40(10), pp. 3258–3263.

Bennati L., Vergara C., Domanin M., Malloggi C., Bissacco D., Trimarchi S., Silani V., Parati G., Casana R., A computational fluid structure interaction study for carotids with different atherosclerotic plaques. Journal of Biomechanical Engineering, 143(9), 091002, 2021

Moreover, in the Discussion, they should compare their results with those of such papers, in terms of confirming or not the outcomes found there

4. A major limitation of the paper is the absence of any turbulence model, which is relevant for stenotic carotids. The authors should emphasize this choice in the methods, include a comment on this in the Discussion/Limitations and add some references, e.g.

Lee S, Lee S, Fischer P, Bassiouny H, Loth F. Direct numerical simulation
of transitional flow in a stenosed carotid bifurcation. J Biomech.
2008;41(11):2551–2561.

Lancellotti R.M.. Vergara C., Valdettaro L., Bose S., Quarteroni A., Large Eddy Simulations for blood fluid-dynamics in real stenotic carotids. Int. J. Numer. Meth. Biomed. Eng., 33(11), e2868, 2017

5. A major issue in FSI for stenotic carotids is the calibration of parameters, in particular elastic properties of the plaque. The authors should mention this issue with suitable references, e.g

Pozzi S., Domanin M., Forzenigo L., Votta E., Zunino P., Redaelli A., Vergara C., A surrogate model for plaque modeling in carotids based on Robin conditions calibrated by cine MRI data . Int. J. Num. Meth. Biomed. Eng., 37(5), e3447, 2021

and detail how did they manage the issue of the choice of the structural parameters, in particular to characterize the three different plaque typologies

6. Sect 2.4: Which boundary condition has been prescribed on the external vessel wall surface?

7. Fig 1: What is the origin of this data? Literature? Please explain and in case cite

8. More details should be provided about FSI numerical coupling method, linearization, time discretization, dt, mesh convergence,....

9. Sect. 2.5: Here also definition of principal stresses (Fig 6) should be reported

Round 2

Reviewer 1 Report

All comments were addressed

Author Response

We would like to thank the reviewer for the comments.

Reviewer 2 Report

Some points need to be fixed:

3. The sentence in the text at line 60 is not in agreement with the message that previous studies have in fact addressed FSI:

"However, the interaction between the blood domain and deforming arterial walls is not considered, where CFD simulations are restricted only to the blood flow analysis. Fluid-structure interaction (FSI) models, combining CFD with structural finite element analysis (FEA), could provide more accurate estimation of real vascular system and have been used to assess both fluid-dynamic and structural behaviors in human atherosclerotic carotid plaques [28–30]."

It should be reformulated, e.g., as follows:

"The interaction between the blood domain and deforming arterial walls is not so much considered, and CFD simulations are ususally restricted only to the blood flow analysis. However, fluid-structure interaction (FSI) models, combining CFD with structural finite element analysis (FEA), have been provided to give more accurate estimation of real vascular system and have been used to assess both fluid-dynamic and structural behaviors in human atherosclerotic carotid plaques [28–30]."

I could not find any discussion about comparison with such studies at line 228.

4. The authors should not think as in a straight cylinder with non pulsatile conditions: for complex geometries and for a pulsatile input, transition to turbulence has been observed in stenotic carotids, independently of the value of Re which holds true in the cylindrical/steady case. It is important that the authors follows my remark. I am not stating to include a turbulence model in tour study, but to add the comments and references as suggested. 

5. For completeness, it is important to mention that some studies used, instead of values taken from the literature, a calibration procedure obtained from clinical images. The authors should iintroduce a comment on this alternative strategy and provide suitable references (e.g. the one reported in the previous report)

6. Ok. But it is important to add such infomation also in the text

Round 3

Reviewer 2 Report

Point 4:

Blood flow through stenotic carotid bifurcations may change to turbulent,

->

Blood flow through stenotic carotid bifurcations may experience transtion to turbulence,

REF [56}: Journal is missing

Author Response

We would like to thank the reviewer for the comments.

The sentence has been changed in revised text (Line 283).

Journal information has been added in REF [56}.